# The Importance of Including Spatial Autocorrelation When Modelling Species Richness in Archipelagos: A Bayesian Approach

Diogo Duarte Barros [1,*], Maria da Luz Mathias [1], Paulo A. V. Borges [2] and Luís Borda-de-Água [3,4,5,*]

1   CESAM—Centro de Estudos do Ambiente e do Mar, Departamento de Biologia Animal, Faculdade de Ciências, Universidade de Lisboa, Edifício C2, 3º piso, Campo Grande, 1749-016 Lisboa, Portugal
2   cE3c—Centre for Ecology, Evolution and Environmental Changes, Azorean Biodiversity Group, CHANGE—Global Change and Sustainability Institute, Faculty of Agricultural Sciences and Environment, University of the Azores, 9700-042 Angra do Heroísmo, Portugal
3   CIBIO/InBio, Centro de Investigação em Biodiversidade e Recursos Genéticos, Laboratório Associado, Universidade do Porto, Campus Agrário de Vairão, 4485-661 Vairão, Portugal
4   CIBIO/InBio, Centro de Investigação em Biodiversidade e Recursos Genéticos, Laboratório Associado, Instituto Superior de Agronomia, Universidade de Lisboa, Tapada da Ajuda, 1349-017 Lisboa, Portugal
5   BIOPOLIS Program in Genomics, Biodiversity and Land Planning, CIBIO, Campus de Vairão, 4485-661 Vairão, Portugal
*   Correspondence: diogo.duarte.barros@gmail.com (D.D.B.); lbagua@gmail.com (L.B.-d.-Á.)

**Abstract:** One of the aims of island biogeography theory is to explain the number of species in an archipelago. Traditionally, the variables used to explain the species richness on an island are its area and distance to the mainland. However, increasing evidence suggests that accounting for other variables is essential for better estimates. In particular, the distance between islands should play a role in determining species richness. This work uses a Bayesian framework using Gaussian processes to assess whether distance to neighbouring islands (spatial autocorrelation) can better explain arthropod species richness patterns in the Azores Archipelago and in the Canary Islands. This method is flexible and allows the inclusion of other variables, such as maximum altitude above sea level (elevation). The results show that accounting for spatial autocorrelation provides the best results for both archipelagos, but overall, spatial autocorrelation seems to be more important in the Canary archipelago. Similarly, elevation plays a more important role in determining species richness in the Canary Islands. We recommend that spatial autocorrelation should always be considered when modelling an archipelago's species richness.

**Keywords:** Azores; biodiversity; biogeography; Canary Islands; elevation; Gaussian process; island species–area relationship; spatial autocorrelation



## 1. Introduction

Islands and archipelagos have provided fertile ground to develop theories in ecology, evolution, and biogeography (e.g., [1–5]). Assessing species richness on islands has been one of the major topics addressed by island biogeographers and one that has had a major impact on the development of theories in ecology overall. One of the best-known theories to explain the patterns and processes determining species richness in islands is the Equilibrium Theory of Island Biogeography by MacArthur and Wilson [6,7], hereafter ETIB. ETIB explains the species richness in an island based on immigration (dispersal) and extinction. In turn, these two processes are a function of the area of the island and its distance to the mainland, the latter acting as a source pool. According to this theory, assuming all other factors (such as climate or topographic complexity) are equal, for two islands at the same distance from the mainland, the largest island has more species, because more migrants are able to reach it (target effect), and there are fewer extinctions (resource availability effect).

Similarly, for islands of the same size, the one closer to the mainland has more species because the number of individuals, and thus of species, reaching it from the mainland is larger and there is also a higher probability of rescue effects.

The pioneering work of MacArthur and Wilson inspired multiple lines of research on biogeography and biodiversity, gaining both support and criticism. For instance, the original ETIB model was criticized for being incomplete because it did not consider the evolutionary time scale on oceanic islands. Such criticisms eventually led to the development of the General Dynamic Model [8], hereafter GDM. In the GDM, the biogeographical patterns and processes of oceanic islands are fundamentally related to the age of the island and its geological evolution. GDM helped to unravel the prominent role of single island endemics, hereafter SIE, as potential indicators of evolutionary dynamics in ocean island archipelagos [2,9,10], by suggesting that the number of SIE on ocean islands is strongly determined by the island's geological age, as reviewed by Borregaard et al. [5]. Moreover, islands with volcanic origin have complex topographies, and maximum altitude above sea level (elevation) can be considered a surrogate of an island's habitat diversity [11] and island speciation rates [12]. The geological and biological history of each island's mountains influences its species richness, level of endemism, and biological dynamics [8]. Therefore, we should expect a high number of SIE in mountainous islands with complex topographies.

The island species–area relationship, hereafter ISAR, relates the number of species, $S$, on an island of area, $A$, and it is often expressed as a power law,

$$S = cA^z, \tag{1}$$

where $c$ and $z$ are constants. This formula has been a favourite among theoreticians and has also received empirical support (e.g., [13,14]). Diamond [15] added an exponential term to describe the effect of the distance of the island to the mainland, $D$,

$$S = c \exp(-dD)A^z, \tag{2}$$

where $d$ is a positive constant (for more recent developments see [1]). However, often islands do not occur in isolation but in archipelagos. It is, then, to be expected that the species richness on an island is not only influenced by its area and distance to the mainland but also by the distance to the other islands in the archipelago [16]. For instance, the distance among islands is likely to affect the ability of species to disperse within the archipelago, which may affect not only species richness but also species composition (and beta-diversity) [16–18]. The latter is not the object of this work; however, closeness may affect species richness because it is likely to be a proxy for similar environmental conditions and for the islands' age [16]. Moreover, some islands may have been connected during glacial periods, which is more likely to have occurred for nearby islands, and their present communities may still reflect a previous species' richer community [2]). Therefore, when trying to fit an ISAR, the distance among islands should be incorporated into the model.

Spatial autocorrelation is often disregarded when fitting the ISAR (e.g., [15]). However, failing to address the effects of spatial autocorrelation may lead, for example, to bias in the parameter estimates. There are several approaches to deal with spatial autocorrelation, such as the use of spatial linear models of the Conditional Autoregressive Model [19], frequently used to address first-order dependencies, or the Simultaneous Autoregressive Model [17], used when there are multiple order dependencies. Dormann et al. [20] extensively reviewed and proposed multiple approaches to deal with spatial autocorrelation, including spatial generalized linear mixed models, where the spatial autocorrelation is modelled by specifying the correlation structure for the residuals. For instance, Selmi and Boulinier [18] suggested the use of spherical, Gaussian, and exponential covariance models to account for the role of spatial covariance in regression models.

Here, we use a model developed by McElreath [21]. It uses a Gaussian process regression (or simply a Gaussian process) to explicitly incorporate spatial autocorrelation. A Gaussian process is a Bayesian method that adds a varying intercept that accounts for the

non-independence of the data, and a multivariate prior for these intercepts [21,22]. This approach has seen multiple practical applications in ecology, namely, within species distribution models [23], inference of function-valued traits [24], and beta diversity predictions as functions of environmental distance [25].

In this work, we illustrate how to model an ISAR with a Gaussian process, thus considering spatial autocorrelation, using data on arthropod species richness from the Azores [26], an archipelago consisting of nine islands, and from the Canary Islands [27], an archipelago consisting of seven islands. In addition to the area of the islands, their distances to the mainland, and the distances between them, we will also consider more complicated models that include the elevation of the islands. Finally, because ISARs only deal with the number of species richness, we will only assess spatial autocorrelation in the context of species richness; for works on community (dis)similarity as a function of distance, see, e.g., [28,29].

## 2. Materials and Methods

### 2.1. Statistical Model

Bayesian methods require that we assign a likelihood for the data and choose priors for the parameters. To choose the likelihood, first notice that the number of species on island $i$, $S_i$, is a discrete number. Two distributions are obvious candidates: the Poisson distribution, which has only one parameter, $\mu$, the mean (which is equal to the variance),

$$S_i \propto \text{Poisson}(\mu_i) \tag{3}$$

and the negative binomial, which has two parameters, $\mu$, the mean, and $\phi$ that controls for over-dispersion,

$$S_i \propto \text{Negative\_binomial}(\mu_i, \phi_i). \tag{4}$$

Typically, the number of species is over-dispersed, that is, the variance is much larger than the mean. This seems to rule out the Poisson distribution; however, this concern is only valid when we do not include a Gaussian process. When we model the parameter $\mu$ of the Poisson distribution with a Gaussian process, the result is a mixture distribution, very much like the negative binomial; recall that a negative binomial is a mixture distribution where the mean of a Poisson distribution is distributed according to a gamma distribution (e.g., [22]). Therefore, for models without the Gaussian process, we use only the negative binomial distribution, but for models with a Gaussian process, we assess both the Poisson and the negative binomial distributions.

As usual, we model the mean, $\mu_i$, as a function of predictor variables, $x_{ij}$. Here we assume power–law relationships,

$$\mu_i = \beta_0 \cdot x_{i1}^{\beta_1} \cdot x_{i2}^{\beta_2} \cdots x_{ij}^{\beta_j}, \tag{5}$$

where $\beta_j$ are the parameters to be estimated. We chose power laws because this is the form often assumed to be the relationship between the area of an island and the number of species, Equation (1) (e.g., [30]). Nevertheless, as we will see later, we also assume an exponential decay for the distance to the mainland.

As mentioned previously, the ISAR is likely to be also determined by factors that are related to the distance between the islands, such as dispersal or similar characteristics shared due to proximity, implying that the data from different islands are not independent. Often, dependence is dealt with by considering multilevel/hierarchical models (e.g., [31,32]). In the case of an ISAR, this would consist of adding a varying intercept, $\gamma_{[i]}$, to Equation (5) (the square brackets emphasize the hierarchical nature of this parameter), and then assuming that the terms $\gamma_{[i]}$ come from a common distribution (e.g., normal), whose parameters

would also need to be estimated. To preclude $\mu_i$ to become negative, $\gamma_{[i]}$ enters Equation (5) as a multiplicative term (e.g., [21]),

$$\mu_i = \beta_0 \cdot e^{\gamma_{[i]}} x_{i1}^{\beta_1} \cdot x_{i2}^{\beta_2} \cdots x_{ij}^{\beta_j}. \tag{6}$$

This approach allows the varying intercepts, $\gamma_{[i]}$, to "absorb" some of the variation associated with each island. However, it assumes that $\gamma_{[i]}$ are discrete categories; therefore, it ignores that spatial autocorrelation is a function of distance and, thus, should be a continuous variable. One way to handle cases where the varying intercepts, $\gamma_{[i]}$, exhibit a continuous dependence is to use a Gaussian Process (e.g., [22]). Specifically, we assume that the varying intercepts $\gamma_{[i]}$ come from a multivariate normal distribution (MVNormal)

$$\gamma \sim \text{MVNormal}(\mathbf{0}, \mathbf{K}),$$

where $\mathbf{0}$ stands for the vector of the means, all equal to zero, and $\mathbf{K}$ is a covariance matrix. The dependence of $\gamma_{[i]}$ on distance is established through the functional form of the covariance matrix $\mathbf{K}$. Often, $\gamma_{[i]}$ is assumed to decay exponentially with the square of the distance (e.g., [21]), with the elements of $\mathbf{K}$, $k_{ij}$, given by

$$k_{ij} = \eta^2 \exp\left(-\rho^2 D_{ij}^2\right) + \delta_{ij}\sigma^2, \tag{7}$$

where $D_{ij}$ stands for the distance between islands $i$ and $j$, and $\eta$, $\rho$, and $\sigma$ are parameters to be estimated. $\delta_{ij}$ is the Kronecker delta, equal to 1 when $i = j$ and zero otherwise, meaning that the term $\delta_{ij}\sigma^2$ corresponds to the variance within an island. However, because there is only one observation per island (its number of species), the term $\delta_{ij}\sigma^2$ is irrelevant (there is no variance among the values for an island); hence, following [21], we set this term equal to a constant (0.01). The above choice of the covariance matrix, Equation (7), is by no means the only possible one; see, for instance, [33]. We tested alternative formulations, but the results were similar.

Notice that the covariance matrix, $\mathbf{K}$, plays a major role in the formulation of the model. It is through this matrix that the distances between the islands, $D_{ij}$, are explicitly included in the model and its elements reflect the importance (or not) of the spatial autocorrelation among the islands. Although this is the matrix whose parameters are being estimated, Equation (7), it is easier to interpret, instead, the correlation matrix (e.g., see [21]). Given two islands, $I$ and $J$, if $k_{i,j}$ is the corresponding element in the covariance and $k_{i,i}$ and $k_{j,j}$ are their variances, then the correlation is calculated using $k_{i,j}/\sqrt{k_{i,i}k_{j,j}}$; naturally, the diagonal elements of the correlation matrix are equal to 1.

In summary, we assume two types of likelihood, Equations (3) and (4), with the means modelled by Equations (5) or (6) (the former only for the negative binomial). This implies that there is one parameter $\beta_0$, $j$ parameters $\beta_j$ (to be identified), the parameter $\phi$ if we used a negative binomial distribution, and $\rho^2$ and $\eta$ when we include the Gaussian process. To illustrate the above model, assume a Poisson distribution with the area of the island, $A_i$, as the only explanatory variable but including a Gaussian process. From Equations (3) and (6), we obtain

$$S_i \sim \text{Poisson}(\mu_i),$$

with

$$\mu_i = \beta_0 e^{\gamma_{[i]}} A_i^z,$$

and recover the classical power–law formulation of the ISAR, Equation (1), with $c = \beta_0 e^{\gamma_{[i]}}$, but where $c$ now includes an autocorrelation term.

Finally, a Bayesian approach also requires that each parameter has a prior distribution. However, the choice of priors depends on the specificities of the situation being analyzed, hence, we defer the discussion on the priors until after the case studies are introduced.

### 2.2. Case Study—The Azores and Canary Archipelagos

2.2.1. Study Area

The Azores and the Canary Islands are two of the archipelagos that make up Macaronesia. The Azores is an isolated northern Atlantic archipelago of nine volcanic islands, located roughly between 37° and 40° N and 24° and 31° W (WGS 84 coordinate system). It is situated across the mid-Atlantic Ridge, which separates the western group (Flores and Corvo) from the central (Faial, Pico, São Jorge, Terceira, and Graciosa) and the eastern (São Miguel and Santa Maria) groups, Figure 1a. The climate is temperate oceanic, and its relative humidity often reaches 95% in the high-elevation native semi-tropical evergreen laurel forest, which contributes to small temperature fluctuations throughout the year [34]. The Canary Islands comprise seven volcanic islands located between 27° and 30° N and 13° and 19° W (WGS 84 coordinate system), distancing roughly 1300 km from the Azores. The Canary Islands are divided into two groups: the western one (Tenerife, La Palma, El Hierro, Gomera, and Gran Canaria) and the eastern one (Fuerteventura and Lanzarote), Figure 1b. The climate is subtropical and desert with a great variation in relative humidity [35].

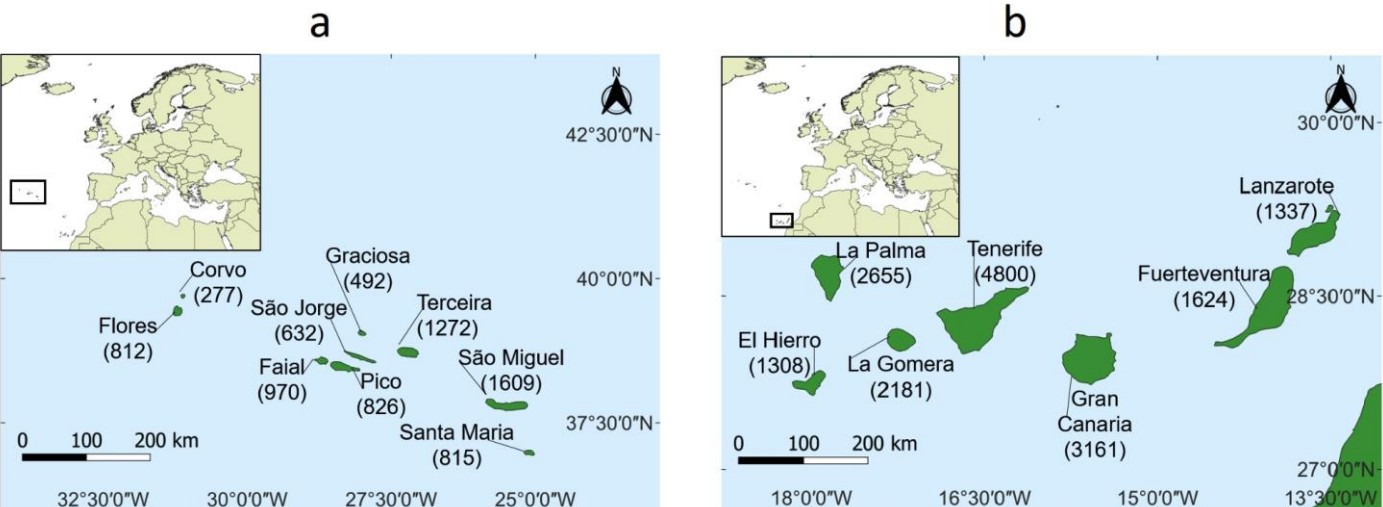

**Figure 1.** The archipelagos of the Azores (**a**) and of the Canary Islands (**b**); the total number of species in each island is shown in brackets.

For each island, we compiled information on the area, geological age, maximum altitude above sea level (elevation) and distance to the mainland [36–42]; see also Tables S1 and S2 in Supplementary Material S1). The distances between the islands were estimated using the Haversine formula and the centroid of each island; see Tables S3 and S4 in Supplementary Material S1.

2.2.2. Arthropod Data

Although new species are added and described occasionally, we believe the current knowledge of the arthropod diversity of these two archipelagos is reasonably complete. We used data from the two most recently updated checklists. For the Azores, this is an updated version of [26] based on new additions from the literature, leading to a total of 2323 species; see Table S5. For the Canary archipelago, we used [27], which contains a total of 7402 species; see Table S6. Therefore, all calculated richness values for each island are for the sums of the lowest taxonomic rank (i.e., species or subspecies). Species were classified as natives (those that were deliberately, or not, introduced by humans) archipelago endemics, single island endemics (those endemic species that occur only on one island, SIE), and multiple island endemics (endemic species that occur on two or more islands, hereafter MIE).

Since the number of endemic species may be determined by processes different from those of the exotic and the native non-endemic species, we modelled separately the total number of species, the number of endemic species, the number of SIE, and the number of MIE species.

### 2.3. Selection of Explanatory Variables

We started by considering the following variables as determinants of species richness: area of the island, age of the island, distance to the mainland, and maximum altitude, i.e., elevation [2,8]. However, the age of an island and the distance to the mainland are correlated; see Figures S1 and S2 in Supplementary Material (a relationship that was stronger for the Canary Islands). This happens because islands are mainly formed along the mid-ocean Atlantic ridge and, as it widens, younger islands are farther away from the continents. Since we have more confidence in the value of the distance to the mainland than in the ages of the islands, we used the former. Thus, in summary, we used "area", "elevation", and "distance to the mainland" as explanatory variables in Equations (5) and (6).

Both archipelagos have an east–west orientation (see Figure 1) and the distance of the islands from the mainland varies greatly within the archipelagos. In the Azores, the island closest to the mainland (Europe) is São Miguel, at a distance of 1368 km, and the farthest is Flores, at 1864 km. In the Canary Islands, the island closest to the mainland (Africa) is Fuerteventura, at a distance of 95 km, and the farthest is La Palma, at 414 km. Some authors (e.g., [15]) have assumed that $S_i$ exhibits an exponential relationship with the distance to the mainland, expression (2). Therefore, in addition to expressions (5) and (6), we also considered:

$$\mu_i = \beta_0 \cdot e^{\gamma_{[i]}} x_{i1}^{\beta_1} \cdot x_{i2}^{\beta_2} \cdots x_{ij}^{\beta_j} e^{-dD_i},$$

where $D_i$ is the distance to the mainland of the island $i$ and $d$ is a parameter to be estimated.

### 2.4. Choice of Priors

When choosing the priors, we aimed at priors that were weakly regularizing, so that they were non-informative for the range of values of interest but allowed for efficient numerical estimation of the posterior distribution [21]. Furthermore, we checked whether the priors led to values that were within our expectations for the number of species in the archipelagos. We adopted the following priors for both archipelagos:

$$\beta_0 \sim \text{normal}(100, 20),$$

$$\beta_j \sim \text{exponential}(1),$$

$$d \sim \text{exponential}(1),$$

$$\rho^2 \sim \text{exponential}(0.5),$$

and

$$\eta \sim \text{exponential}(1).$$

### 2.5. Software Packages

All the analyses were performed in R version 4.2.0 [43] with the packages Rethinking [21] and RStan [44]. We checked the convergence of the models obtained by confirming that the $\hat{R}$ ("rhat", see Supplementary Information S2) was not greater than 1. $\hat{R}$ measures variance within all MCMC chain samples, and if all chains are at equilibrium, they should have the same variance, thus, $\hat{R} = 1$ [44]. In addition, we also show the effective number of samples, "n_eff", that is, the estimated number of independent samples [44].

## 3. Results

### 3.1. The Traditional ISAR with and without the GAUSSIAN Process

Because the traditional power–law form of the ISAR, $S = cA^z$, has played an important role in ecological theory (e.g., [6]), our analyses start with this version with and without the Gaussian process. The main purpose of this analysis is to illustrate the advantage of including a Gaussian process. For ease of comparison, we use in both cases the negative binomial likelihood. Tables 1 and 2 show the values of the parameters and Figure 2a,b show the fitting.

**Table 1.** Stan output for the ISAR model with and without the Gaussian process (GP) for the Azores. "SD" stands for standard deviation, "CV" for coefficient of variation, "2.50%" and "97.50%" correspond to the width of the credible interval, "n_eff" is the effective sample size, and "Rhat" stands for $\hat{R}$ (see main text).

| | | Mean | SD | CV | 2.50% | 97.50% | n_eff | Rhat |
|---|---|---|---|---|---|---|---|---|
| Without GP | $c$ | 100.64 | 20.23 | 0.20 | 63.33 | 139.77 | 223 | 1.00 |
| | $z$ | 0.41 | 0.05 | 0.12 | 0.32 | 0.52 | 172 | 1.00 |
| | $\phi$ | 4.53 | 1.99 | 0.44 | 1.66 | 9.36 | 195 | 1.00 |
| With GP | $c$ | 99.53 | 19.16 | 0.19 | 64.45 | 136.28 | 219 | 1.01 |
| | $z$ | 0.42 | 0.06 | 0.14 | 0.31 | 0.56 | 123 | 1.00 |
| | $\phi$ | 6.47 | 3.17 | 0.49 | 2.05 | 14.54 | 232 | 1.02 |
| | $\eta^2$ | 0.23 | 0.48 | 2.09 | 0.00 | 1.39 | 363 | 1.01 |
| | $\rho^2$ | 1.56 | 1.83 | 1.17 | 0.02 | 6.22 | 406 | 1.00 |

**Table 2.** Stan output for the ISAR model with and without the Gaussian process (GP) for the Canary Islands. "SD" stands for standard deviation, "CV" for coefficient of variation, "2.50%" and "97.50%" correspond to the width of the credible interval, "n_eff" is the effective sample size and "Rhat" stands for $\hat{R}$ (see main text).

| | | Mean | SD | CV | 2.50% | 97.50% | n_eff | Rhat |
|---|---|---|---|---|---|---|---|---|
| Without GP | $c$ | 101.91 | 20.31 | 0.20 | 61.69 | 139.74 | 230 | 1.00 |
| | $z$ | 0.47 | 0.05 | 0.11 | 0.38 | 0.59 | 172 | 1.01 |
| | $\phi$ | 2.93 | 1.47 | 0.50 | 0.89 | 6.00 | 224 | 1.00 |
| With GP | $c$ | 99.31 | 20.4 | 0.20 | 57.9 | 136.31 | 248 | 1.00 |
| | $z$ | 0.46 | 0.10 | 0.22 | 0.18 | 0.63 | 54 | 1.00 |
| | $\phi$ | 5.03 | 2.64 | 0.52 | 1.30 | 11.49 | 292 | 1.00 |
| | $\eta^2$ | 0.58 | 0.80 | 1.34 | 0.02 | 2.69 | 122 | 1.00 |
| | $\rho^2$ | 1.37 | 1.96 | 1.43 | 0.02 | 7.79 | 204 | 1.00 |

Interestingly, the means and standard deviations of the $c$ parameters with and without the Gaussian Process are similar for both archipelagos, but the parameters $z$ and $\phi$ have large coefficients of variation in the model with the Gaussian process. This reveals that when spatial autocorrelation is not included, we have undue certainty about some parameters.

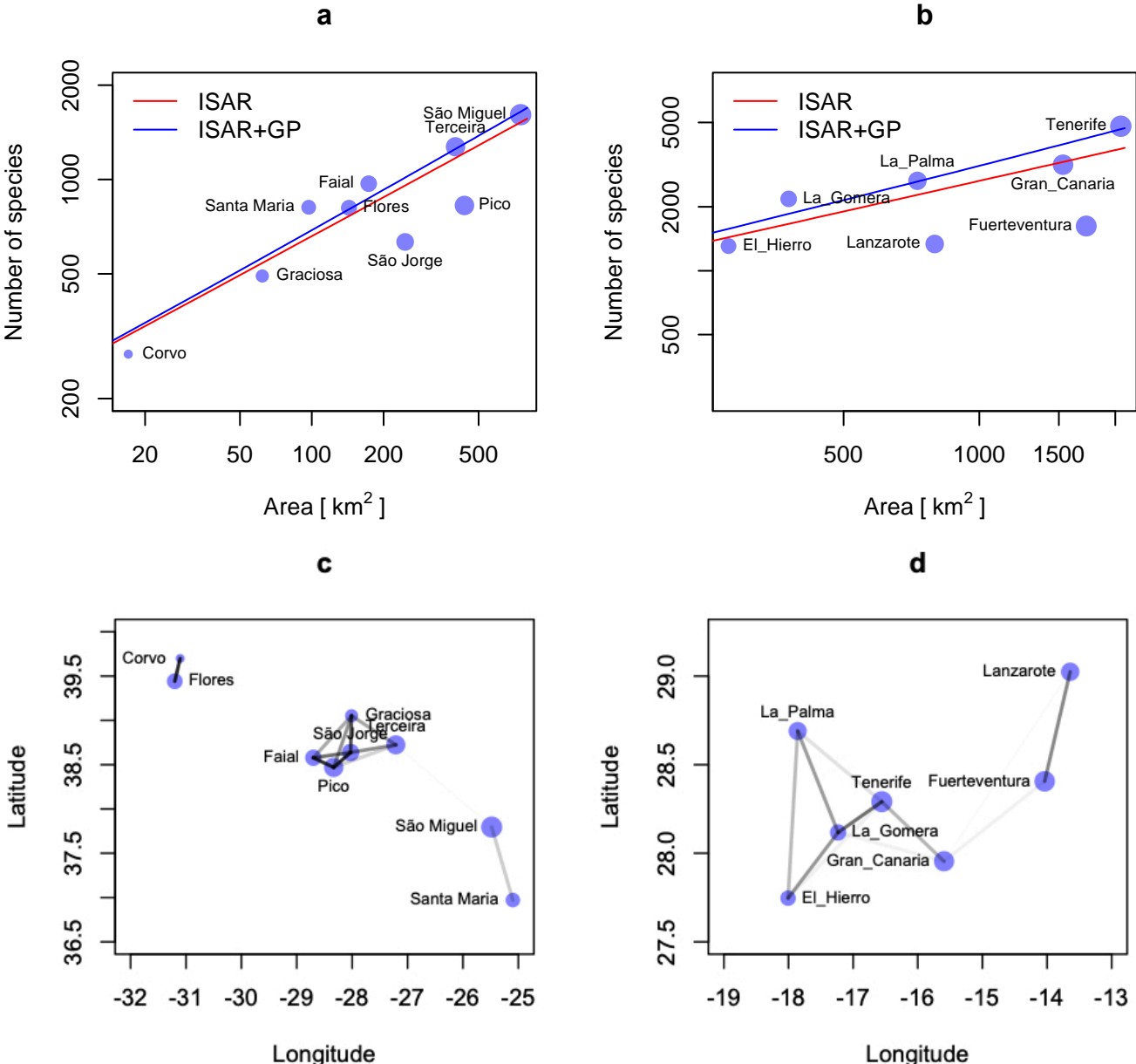

**Figure 2.** Plots (**a**,**b**) show the fitting with the Gaussian process, GP, (blue line) and without (red line) for the island species–area relationship (ISAR) model. The shadow areas correspond to 95% credible intervals. The fitting lines were obtained with the mean values of the posterior of the parameters *c* and *z* of $S = cA^z$. For the model with the Gaussian process, the fitting line corresponds only to $S = cA^z$, i.e., without the Gaussian process term. Plots (**c**,**d**) are geographical locations of the archipelagos and the darkness of the lines among the islands represents the strength of the correlations (for some pairs of islands, the correlation value is so small that the line is not visible). The size of the dots is proportional to the area of the islands.

We used the "Widely Applicable Information Criterion" (WAIC) to compare the fitting of the two models, and adopted the terminology suggested by [45] to compare the performance of the models. Despite the larger number of parameters, WAIC shows that the model with the Gaussian process performed best in both archipelagos; see Table 3. However, the improvement is greater for the Canary Islands than for the Azores. Indeed, as Figure 2a shows, the uncertainty of the fitting provided by the two models is similar for the Azores but very different for the Canary Islands.

**Table 3.** Comparison of ISAR models using the "Widely Applicable Information Criterion" (WAIC). ΔWAIC for model $i$ is calculated as $\text{WAIC}_i$—$\text{WAIC}_{min}$, and "weight" is calculated as $\exp(-0.5\Delta\text{WAIC}_i)/\sum_{\text{All models}}\exp(-0.5\Delta\text{WAIC}_i)$. "GP" stands for Gaussian Process.

| ISAR Model | | WAIC | ΔWAIC | Weight |
|---|---|---|---|---|
| Azores | With GP | 127.00 | 0.00 | 0.56 |
| | Without GP | 127.50 | 0.50 | 0.44 |
| Canary Islands | With GP | 116.70 | 0.00 | 0.76 |
| | Without GP | 119.00 | 2.30 | 0.24 |

Spatial autocorrelation is modelled through the covariance matrix, *K*; however, as we previously mentioned, the correlation matrix is easier to interpret [21]. To obtain the correlation matrix, because both parameters have skewed distributions, we use the median of the $\eta^2$ and $\rho^2$ posteriors. The correlation matrices are shown in Tables 4 and 5 for the Azores and the Canary Islands, respectively. Figure 2c,d shows the archipelagos and the intensity of the color of the line between the islands reflects the value of the correlation; for some islands, the correlation is so weak that the lines are not easily discerned. Although the model does not consider the species' identity, only species richness, visual inspection of Figure 2c,d reveals clusters of islands in the Azores and in the Canary Islands. These clusters are clearly based on proximity among the islands; thus, independently of species identity, species richness alone is highly influenced by proximity.

**Table 4.** Correlation matrix of the ISAR model with a Gaussian process for the Azores (median values). The grey shading areas highlight the three major groups of islands: the western, central, and eastern groups. The abbreviations mean: C—Corvo, Fl—Flores, Fa—Faial, P—Pico, G—Graciosa, SJ—São Jorge, T—Terceira, SM—São Miguel, SMa—Santa Maria.

| | C | Fl | Fa | P | G | SJ | T | SM | SMa |
|---|---|---|---|---|---|---|---|---|---|
| C | 1.000 | 0.831 | 0.004 | 0.001 | 0.001 | 0.000 | 0.000 | 0.000 | 0.000 |
| Fl | 0.831 | 1.000 | 0.005 | 0.001 | 0.001 | 0.000 | 0.000 | 0.000 | 0.000 |
| Fa | 0.004 | 0.005 | 1.000 | 0.810 | 0.499 | 0.654 | 0.185 | 0.000 | 0.000 |
| P | 0.001 | 0.001 | 0.810 | 1.000 | 0.567 | 0.819 | 0.346 | 0.002 | 0.000 |
| G | 0.001 | 0.001 | 0.499 | 0.567 | 1.000 | 0.740 | 0.510 | 0.002 | 0.000 |
| SJ | 0.000 | 0.000 | 0.654 | 0.819 | 0.740 | 1.000 | 0.560 | 0.004 | 0.000 |
| T | 0.000 | 0.000 | 0.185 | 0.346 | 0.510 | 0.560 | 1.000 | 0.040 | 0.001 |
| SM | 0.000 | 0.000 | 0.000 | 0.002 | 0.002 | 0.004 | 0.040 | 1.000 | 0.371 |
| SMa | 0.000 | 0.000 | 0.000 | 0.000 | 0.000 | 0.000 | 0.001 | 0.371 | 1.000 |

**Table 5.** Correlation matrix of the ISAR model with a Gaussian process for the Canary Islands (median values). The grey shading areas highlight the two major groups of islands: the western and eastern groups. The abbreviations mean: F—Fuerteventura, L—Lanzarote, GC—Gran Canaria, T—Tenerife, LG—La Gomera, EH—El Hierro, LP—La Palma.

| | F | L | GC | T | LG | EH | LP |
|---|---|---|---|---|---|---|---|
| F | 1.000 | 0.506 | 0.081 | 0.003 | 0.000 | 0.000 | 0.000 |
| L | 0.506 | 1.000 | 0.008 | 0.000 | 0.000 | 0.000 | 0.000 |
| GC | 0.081 | 0.008 | 1.000 | 0.345 | 0.078 | 0.004 | 0.005 |
| T | 0.003 | 0.000 | 0.345 | 1.000 | 0.579 | 0.097 | 0.166 |
| LG | 0.000 | 0.000 | 0.078 | 0.579 | 1.000 | 0.449 | 0.436 |
| EH | 0.000 | 0.000 | 0.004 | 0.097 | 0.449 | 1.000 | 0.317 |
| LP | 0.000 | 0.000 | 0.005 | 0.166 | 0.436 | 0.317 | 1.000 |

*3.2. Considering All Explanatory Variables*

　　We next considered models with each of the explanatory variables, "area", "elevation", and "distance to the mainland" separately and all together (we call the latter the "full" model). We fitted the models for different types of species: all species, native, endemic, SIE, and MIE species. Tables 6–10 show the results of the WAIC scores for these models; to simplify the presentation, we only show the results for models with WAIC "weights" larger than 0.1; Supplementary Materials S3 and S4 show the full results. These tables reveal that the best models are always the ones with the Gaussian process (despite the increased number of parameters) and Poisson likelihood. (Therefore, in the following, we identify the models solely by their explanatory variables and omit that they include the Gaussian process and that the likelihood is a Poisson distribution).

**Table 6.** Comparison of the several models using the "Widely Applicable Information Criterion" (WAIC) criterion for all species. Only models with 'weight' greater than 0.01 are shown. See the caption of Table 3 for the explanation of the acronyms "ΔWAIC" and "weight". "poisson" means that the model assumes a Poisson likelihood and "gp" means it included a Gaussian process (thus necessarily the distance among islands). "area", "dist" and "elev" stand for the models that (in addition to the Gaussian process) also consider area, distance to the mainland, or elevation, respectively, while "full" stands for a model with all these three explanatory variables included. The "e" in "diste" and "fulle" means that the distance to the mainland entered the model through an exponential relationship, and "dist" and "full" through a power–law relationship.

| Archipelago | Model | WAIC | ΔWAIC | Weight |
|---|---|---|---|---|
| | fulle.poisson.gp | 91.6 | 0.0 | 0.23 |
| | area.poisson.gp | 91.7 | 0.1 | 0.21 |
| Azores | dist.poisson.gp | 91.9 | 0.3 | 0.20 |
| | diste.poisson.gp | 92.0 | 0.4 | 0.19 |
| | elev.poisson.gp | 92.2 | 0.5 | 0.17 |
| | area.poisson.gp | 78.3 | 0.0 | 0.20 |
| | elev.poisson.gp | 78.4 | 0.1 | 0.19 |
| | dist.poisson.gp | 78.5 | 0.3 | 0.17 |
| Canary Islands | fulle.poisson.gp | 78.7 | 0.4 | 0.16 |
| | full.poisson.gp | 78.9 | 0.7 | 0.14 |
| | diste.poisson.gp | 79.1 | 0.8 | 0.13 |

　　For all species (Table 6) and for the Azores, the model with the lowest WAIC was the full one with distance to the mainland modelled by an exponential. However, the ΔWAIC reveals that four other models also have considerable support. For the Canary Islands, the area model had the smallest WAIC, followed by the elevation model, and according to the ΔWAIC, the four other models also have considerable support.

　　For the native species (Table 7) and for the Azores, the model with the lowest WAIC was the elevation-alone model. However, all the other models with the Gaussian process have very similar WAIC, as revealed by the ΔWAIC and the respective "weights". A similar situation occurs for the Canary Islands, but now the model with the smallest WAIC is the "fulle" one, immediately followed by the elevation-alone one.

**Table 7.** Comparison of the several models using the "Widely Applicable Information Criterion" (WAIC) criterion for the native species. Only models with 'weight' greater than 0.01 are shown. See the captions of Tables 3 and 6 for the explanation of the acronyms.

| Archipelago | Model | WAIC | ΔWAIC | Weight |
|---|---|---|---|---|
| Azores | elev.poisson.gp | 85.7 | 0.0 | 0.18 |
| | area.poisson.gp | 85.8 | 0.1 | 0.17 |
| | fulle.poisson.gp | 85.8 | 0.1 | 0.17 |
| | full.poisson.gp | 85.8 | 0.2 | 0.16 |
| | dist.poisson.gp | 85.9 | 0.2 | 0.16 |
| | diste.poisson.gp | 85.9 | 0.2 | 0.16 |
| Canary Islands | fulle.poisson.gp | 62.1 | 0.0 | 0.20 |
| | elev.poisson.gp | 62.2 | 0.1 | 0.19 |
| | area.poisson.gp | 62.4 | 0.3 | 0.17 |
| | full.poisson.gp | 62.4 | 0.3 | 0.17 |
| | dist.poisson.gp | 62.8 | 0.8 | 0.14 |
| | diste.poisson.gp | 62.9 | 0.8 | 0.13 |

**Table 8.** Comparison of the several models using the "Widely Applicable Information Criterion" (WAIC) criterion for the endemic species. Only models with 'weight' greater than 0.01 are shown. See the captions of Tables 3 and 6 for the explanation of the acronyms.

| Archipelago | Model | WAIC | ΔWAIC | Weight |
|---|---|---|---|---|
| Azores | area.poisson.gp | 70.7 | 0.0 | 0.38 |
| | fulle.poisson.gp | 72.2 | 1.5 | 0.18 |
| | full.poisson.gp | 72.8 | 2.1 | 0.13 |
| | dist.poisson.gp | 73.2 | 2.5 | 0.11 |
| | diste.poisson.gp | 73.3 | 2.6 | 0.10 |
| Canary Islands | fulle.poisson.gp | 70.8 | 0.0 | 0.20 |
| | area.poisson.gp | 70.9 | 0.1 | 0.18 |
| | elev.poisson.gp | 71.0 | 0.2 | 0.17 |
| | full.poisson.gp | 71.2 | 0.4 | 0.16 |
| | dist.poisson.gp | 71.3 | 0.5 | 0.15 |
| | diste.poisson.gp | 71.6 | 0.8 | 0.13 |

For the endemic species (Table 8) and for the Azores, the area-alone model has the smallest WAIC. For the Canary Islands, the best is the full model, with distance to the mainland modelled by an exponential, but the area, elevation, and distance models also have substantial support. As before, the models with "elevation" have a smaller ΔWAIC for the Canaries than for the Azores.

For the SIE species (Table 9) and for the Azores, the full model, with distance to the mainland modelled by an exponential, has the smallest WAIC. The second-best model has much less support. As in the previous cases, for the Canary Islands, there are several models with similar WAICs. The model with lower WAIC is the distance to the mainland only, modelled with a power–law decay.

**Table 9.** Comparison of the several models using the "Widely Applicable Information Criterion" (WAIC) criterion for the single island endemic species (SIE). Only models with 'weight' greater than 0.01 are shown. See the captions of Tables 3 and 6 for the explanation of the acronyms.

| Archipelago | Model | WAIC | ΔWAIC | Weight |
|---|---|---|---|---|
| Azores | fulle.poisson.gp | 57.8 | 0.0 | 0.45 |
| | diste.poisson.gp | 59.7 | 2.0 | 0.17 |
| | full.poisson.gp | 60.3 | 2.6 | 0.13 |
| Canary Islands | dist.poisson.gp | 61.3 | 0.0 | 0.19 |
| | elev.poisson.gp | 61.4 | 0.1 | 0.18 |
| | diste.poisson.gp | 61.4 | 0.1 | 0.18 |
| | area.poisson.gp | 61.6 | 0.3 | 0.17 |
| | full.poisson.gp | 61.6 | 0.3 | 0.16 |
| | fulle.poisson.gp | 62.2 | 0.9 | 0.12 |

**Table 10.** Comparison of the several models using the "Widely Applicable Information Criterion" (WAIC) criterion for the multiple island endemic species (MIE). Only models with 'weight' greater than 0.01 are shown. See the captions of Tables 3 and 6 for the explanation of the acronyms.

| Archipelago | Model | WAIC | ΔWAIC | Weight |
|---|---|---|---|---|
| Azores | area.poisson.gp | 69.6 | 0.0 | 0.4.0 |
| | fulle.poisson.gp | 70.2 | 0.6 | 0.29 |
| | dist.poisson.gp | 72.1 | 2.6 | 0.11 |
| | elev.poisson.gp | 72.4 | 2.8 | 0.10 |
| | diste.poisson.gp | 72.4 | 2.9 | 0.10 |
| Canary Islands | elev.poisson.gp | 68.7 | 0.0 | 0.23 |
| | dist.poisson.gp | 68.8 | 0.1 | 0.22 |
| | fulle.poisson.gp | 68.9 | 0.2 | 0.20 |
| | area.poisson.gp | 68.9 | 0.3 | 0.20 |
| | diste.poisson.gp | 69.5 | 0.9 | 0.15 |

Finally, for the MIE species (Table 10) and for the Azores, the models with more support are the area-only and the "fulle", and for the Canaries, the one with smallest WAIC is the elevation-only model, although all the others also show significant support. Again, the ΔWAIC among models is smaller for the Canaries than it is for the Azores. Continuing the previous trend, the model with "elevation" only tends to exhibit lower ΔWAIC for the Canary Islands than for the Azores.

Analysis of the correlation matrices of the best models for the MIE and SIE species further highlights the importance of considering spatial autocorrelation. Because MIE species occur in several islands, dispersal is likely to determine their geographical distributions and proximity between islands should facilitate dispersal. On the other hand, the number of SIE species in an island should be less autocorrelated with the number of SIE in surrounding islands, but it would be incorrect to assume that there is no correlation, because proximity among islands may imply more similar geological ages or environmental conditions that may influence SIE species richness. In any case, we expect MIE species to display higher levels of positive spatial autocorrelation than SIE species. If that is the case, the matrix of correlation of the MIE species should have larger values than that of the SIE species. This is indeed what is observed, a result that becomes more evident if we divide each element of the MIE correlation matrix by its corresponding element in the SIE correlation matrix; see Tables 11 and 12 (see also Tables S7–S10).

**Table 11.** Ratio of the correlation matrices for the best models for MIE and SIE species of the Azores. The abbreviations mean: C—Corvo, Fl—Flores, Fa—Faial, P—Pico, G—Graciosa, SJ—São Jorge, T—Terceira, SM—São Miguel, SMa—Santa Maria.

|  | C | Fl | Fa | P | G | SJ | T | SM | SMa |
|---|---|---|---|---|---|---|---|---|---|
| C | 1.00 | 1.14 | 0.00 | 0.00 | 0.00 | 0.00 | 0.00 | 0.00 | 0.00 |
| Fl | 1.14 | 1.00 | 0.00 | 0.00 | 0.00 | 0.00 | 0.00 | 0.00 | 0.00 |
| Fa | 0.00 | 0.00 | 1.00 | 1.20 | 2.74 | 1.72 | 14.85 | 0.00 | 0.00 |
| P | 0.00 | 0.00 | 1.20 | 1.00 | 2.20 | 1.17 | 5.13 | 0.00 | 0.00 |
| G | 0.00 | 0.00 | 2.74 | 2.20 | 1.00 | 1.40 | 2.63 | 0.00 | 0.00 |
| SJ | 0.00 | 0.00 | 1.72 | 1.17 | 1.40 | 1.00 | 2.25 | 0.00 | 0.00 |
| T | 0.00 | 0.00 | 14.85 | 5.13 | 2.63 | 2.25 | 1.00 | 204.25 | 0.00 |
| SM | 0.00 | 0.00 | 0.00 | 0.00 | 0.00 | 0.00 | 204.25 | 1.00 | 4.54 |
| SMa | 0.00 | 0.00 | 0.00 | 0.00 | 0.00 | 0.00 | 0.00 | 4.54 | 1.00 |

**Table 12.** Ratio of the correlation matrices for the best models for MIE and SIE species of the Canary Islands. The abbreviations mean: F—Fuerteventura, L—Lanzarote, GC—Gran Canaria, T—Tenerife, LG—La Gomera, EH—El Hierro, LP—La Palma.

|  | F | L | GC | T | LG | EH | LP |
|---|---|---|---|---|---|---|---|
| F | 1.00 | 1.44 | 5.08 | 47.14 | 0.00 | 0.00 | 0.00 |
| L | 1.44 | 1.00 | 24.88 | 0.00 | 0.00 | 0.00 | 0.00 |
| GC | 5.08 | 24.88 | 1.00 | 1.88 | 5.22 | 37.23 | 35.4 |
| T | 47.14 | 0.00 | 1.88 | 1.00 | 1.31 | 4.51 | 3.11 |
| LG | 0.00 | 0.00 | 5.22 | 1.31 | 1.00 | 1.56 | 1.60 |
| EH | 0.00 | 0.00 | 37.23 | 4.51 | 1.56 | 1.00 | 1.99 |
| LP | 0.00 | 0.00 | 35.4 | 3.11 | 1.6 | 1.99 | 1.00 |

## 4. Discussion

The biogeographical processes that shape the species richness of an archipelago are complex and disentangling the effects of the driving factors behind these processes is challenging [1,4]. While island area and distance to the mainland are well-studied driving factors, others, such as the distance between islands within an archipelago, are not. Although dispersal between islands is widely recognized as extremely important for determining their species richness [16,46,47], the original ETIB only considered the distance between islands and the mainland. The novelty of our work, in the context of species richness in island biogeography, consists of also including the distance between the islands, and hence considering spatial autocorrelation using a Gaussian process (see also [48]). Our analyses demonstrated that spatial autocorrelation should not be left unaddressed in island biogeography studies for at least the following four reasons: (i) it affects the estimation of the parameters (e.g., their standard errors), (ii) reveals characteristics of the archipelagos (e.g., clustering of islands), (iii) helps identify differences between different types of richness measures (SIE versus MIE), and (iv) reveals differences between archipelagos (those where spatial autocorrelation plays a more important role). Indeed, ignoring spatial autocorrelation can eventually lead to wrong conclusions and even opposite patterns [17,49].

The formulation adopted [21] allows for the inclusion of several variables. In fact, although important, island area and distances to the mainland or between islands are not the only determinants of species richness in an archipelago. Under the GDM [8, 50], the age of an island determines its topographic complexity and area, that is, the

latter are not constant over time. There are no data to assess the temporal dynamics of species richness in the Azores and Canary archipelagos at the time scales relevant for the GDM. However, an important insight from GDM theory is that topographical complexity determines species richness. In fact, there are a large number of theories arguing for a different range of determinants of species richness as a function of topographical isolation and elevation ([12,51–53]), with some arguing that "diversity begets diversity" [10] (but see [54]). Our work may add to this debate by providing a method to include spatial autocorrelation. Overall, elevation seems to have a more dominant role in the Canary Islands than it has in the Azores. This may happen because the average maximum altitude is higher in the Canary Islands (~1800 m) than in the Azores (~1000 m) and because, even for the same range of altitudes, the climatic conditions are more homogeneous in the Azores [13].

An additional advantage of including spatial autocorrelation using a Gaussian process was the identification of island clusters within archipelagos through the correlation matrix [21]. We suggest that this method can be used routinely in the future to identify clusters. The correlation matrices also allowed the identification of the relative importance of spatial autocorrelation for SIE and MIE species richness. As expected, the correlation among islands is higher for MIE species, likely reflecting the importance of dispersal. Nevertheless, the correlation was different from zero among some islands for SIE species, probably due to similar environmental conditions and histories among nearby islands.

Finally, note that our approach only dealt with the number of species, not with their identities. Future work assessing the degree of community similarity as a function of distance is also possible within a Bayesian framework [28,29] and we plan to address this in future work. In summary, we developed a methodology to estimate species richness in an archipelago using Bayesian methods, paying special attention to the spatial autocorrelation, i.e., the influence of the distance between islands, which we modelled using a Gaussian process. We applied our methodology to the Azores and Canary Islands arthropod diversity, in two archipelagos of Macaronesia, and our approach highlighted the differences between them. The existence of off-the-shelf methods to identify spatial autocorrelations, as our results showed, warrants the conclusion that spatial autocorrelation should always be initially considered when studying the species richness of archipelagos, even if this analysis ends up showing that it is not present, which in itself would be an important conclusion.

**Supplementary Materials:** The following supporting information can be downloaded at: https://www.mdpi.com/article/10.3390/d15020127/s1, Information on the geographical and geological characteristics, and species richness, of the islands of the Azores and Canary archipelagos: Supplementary_Material_S1.pdf; selection of the explanatory variables: Supplementary_Material_S2.pdf; results for the models with the lowest WAIC: Supplementary_Material_S3.pdf; results for all models: Supplementary_Material_S4.pdf; R and Stan codes: Supplementary_Material_S5.zip.

**Author Contributions:** D.D.B. and L.B.-d.-Á. developed the Bayesian modelling framework and analysed the model output. P.A.V.B. was responsible for data collection and curation. M.d.L.M. provided important expertise in biogeography and ecology. D.D.B. and L.B.-d.-Á. wrote the first draft of the manuscript, while all authors contributed substantially to the revisions. All authors have read and agreed to the published version of the manuscript.

**Funding:** D.B. was funded by an FCT grant as part of the Biology and Ecology of Global Changes doctoral program. L.B.A. was financed under the Norma Transitória—L57/2016/CP1440/CT0022. P.A.V.B. worked on the manuscript under the framework of the Project FCT—Fundação para a Ciência e a Tecnologia, I.P., under the project UIDP/05292/2020 and UIDB/05292/2020. M.L.M. was supported by FCT/MCTES (UIDP/50017/2020 + UIDB/50017/2020 + LA/P/0094/2020).

**Data Availability Statement:** All the data used in this work are listed in Supplementary Material S1.

**Acknowledgments:** We thank Michael Betancourt, Giovani Silva, and Filipe Dias for their discussions on Bayesian theory and its implementation.

**Conflicts of Interest:** The authors declare no conflict of interest.

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
