# Peer review of "The Importance of Including Spatial Autocorrelation When Modelling Species Richness in Archipelagos: A Bayesian Approach"

_diversity, doi:10.3390/d15020127_

Round 1

Reviewer 1 Report

This is an interesting study contributing to the classic question on how to explain richness patterns on oceanic islands. Authors provide data from a large number of species of a very diverse group (Coleoptera) from two archipelagos and test the importance of considering distance among islands in the models to predict species numbers. The manuscript is well-written and results are somewhat clear. However, there is still room for improvement as somethings need to be clarified. For example, some sentences in the results, and subsequently in the discussion, need to be rewritten to show that some models have no significant differences from others (see detailed comments). Also some questions need to be addressed: how the taxonomic resolution used in this study may bias the results? Was the taxonomy updated before performing the models? Would they be different if considering subspecies as well? Did authors test this? Also, I would exclude exotics from the total number of species, as their colonization processes are very different from the native ones. Authors should recalculate the results taking out introduced taxa not to bias results of total number of species and discuss the difference of including them in results. Did the area of all the islands of both archipelagos kept the same since they emerged above sea-level? Or some were connected during some periods of time?

Some parts of the manuscript need to be restructured (e.g., some information provided in the results is, in fact, methodology description)

In addition, figures need to be substantially improved. Captions need to be self-explanatory. If not, Authors should clearly state were readers can find the information referred to in it.

Finally, some references are lacking to be included in the discussion to show that distance among island was as already considered in other phylogeography works, including for the Canary Islands (e.g., Sanmartín & Ronquist 2008) as well as taxa richness works as function of habitat, distance and/or elevation following the GDM has also been proven for other Macaronesian Archipelagos, for other arthropods groups and flora and vertebrate groups. This will provide support to this study and broaden the discussion, making it more interesting for a wider range of readers.

Please find my detailed comments to the authors in the PDF file marked with comments and track changes. I sincerely hope those are useful for the authors to improve the manuscript.

Minor comments:

Exclude details from abstracts.

In the introduction, please refer to the formulas of models of the latest papers on specific models for oceanic islands that also take into account the age of the islands (e.g., Whittaker et al 2008 and following articles).

Merge Tables 1 with 2 and Tables 6 – 9 as they repeat the same information. Use the same number of digits for all figures.

In the tables and figures, please detail in the caption the meaning of all acronyms used.

Author Response

Reviewer 1

This is an interesting study contributing to the classic question on how to explain richness patterns on oceanic islands. Authors provide data from a large number of species of a very diverse group (Coleoptera) from two archipelagos and test the importance of considering distance among islands in the models to predict species numbers. The manuscript is well-written and results are somewhat clear. However, there is still room for improvement as somethings need to be clarified. For example, some sentences in the results, and subsequently in the discussion, need to be rewritten to show that some models have no significant differences from others (see detailed comments). Also some questions need to be addressed: how the taxonomic resolution used in this study may bias the results? Was the taxonomy updated before performing the models? Would they be different if considering subspecies as well? Did authors test this? Also, I would exclude exotics from the total number of species, as their colonization processes are very different from the native ones. Authors should recalculate the results taking out introduced taxa not to bias results of total number of species and discuss the difference of including them in results. Did the area of all the islands of both archipelagos kept the same since they emerged above sea-level? Or some were connected during some periods of time? 

Some parts of the manuscript need to be restructured (e.g., some information provided in the results is, in fact, methodology description) 

In addition, figures need to be substantially improved. Captions need to be self-explanatory. If not, Authors should clearly state were readers can find the information referred to in it.

Finally, some references are lacking to be included in the discussion to show that distance among island was as already considered in other phylogeography works, including for the Canary Islands (e.g., Sanmartín & Ronquist 2008) as well as taxa richness works as function of habitat, distance and/or elevation following the GDM has also been proven for other Macaronesian Archipelagos, for other arthropods groups and flora and vertebrate groups. This will provide support to this study and broaden the discussion, making it more interesting for a wider range of readers.

Please find my detailed comments to the authors in the PDF file marked with comments and track changes. I sincerely hope those are useful for the authors to improve the manuscript.

Minor comments: 

Exclude details from abstracts.

In the introduction, please refer to the formulas of models of the latest papers on specific models for oceanic islands that also take into account the age of the islands (e.g., Whittaker et al 2008 and following articles).

Merge Tables 1 with 2 and Tables 6 – 9 as they repeat the same information. Use the same number of digits for all figures. 

In the tables and figures, please detail in the caption the meaning of all acronyms used.

We thank the reviewer for the careful reading of the manuscript and for her/his suggestions that will definitely improve the manuscript, though we do not agree with all of them (mainly minor details). We believe we answered all the comments raised above below, but one deserves special attention:

“I would exclude exotics from the total number of species, as their colonization processes are very different from the native ones. Authors should recalculate the results taking out introduced taxa not to bias results of total number of species and discuss the difference of including them in results.”

We do not fully agree with this statement. First, it is assumed that the colonization process is different for exotic species. Although this is likely to be the case for some, such as the invasive ones, it is not clear that it is necessarily the case for all exotic species. This may be especially the case among such a rich group as the arthropods and our present lack of detailed knowledge about every single species. Second, we believe it is important to show also the results to the entire community.

However, we do agree with the referee that it is worth also showing results for the native species. Therefore, we changed subsection 2.2.2 reflecting that we also now add the category of “natives” and made other changes in the text. For example

“We fitted the models for different types of species: all species, native, endemic, SIE and MIE species” instead of  “We fitted the models for different types of species: all species, endemic, SIE and MIE species”

or “Tables 6 to 10”, instead of “Tables 6 to 9” because we now have another table to the new species group considered.

We added the following paragraph to the Results:

“For the native species (Table 7) and for the Azores, the model with the lowest WAIC was the elevation-alone model. However, all the other models with the Gaussian process have very similar WAIC, as revealed by the DWAIC and the respective “weights”. A similar situation occurs for the Canary Islands, but now the model with the smallest WAIC is the “fulle” one, immediately followed by the elevation alone one.”

Line 24 – Abstract – “please rephrase to 'in the Azores Archipelago and in the Canary Islands' or something similar (weird phrasing)

Thank you for this suggestion. We changed as recommended

Line 26 – Abstract – “exclude details from abstracts or just merge this idea with the sentence in line 23 by referring that 'several hundreds of arthropod species' after 'framework using'

We removed the sentence: “We studied 2323 species of arthropods from the Azores and 7402 from the Canary Islands.”

Line 70 – “Please check the latest papers on specific models for oceanic islands that also take into account the age of the islands (e.g., Whittaker et al 2008 and following articles) and refer to those instead

We always like to provide references to the original papers. The paper by Whittaker et al 2008 addresses mainly the temporal component, their ATT2 model, and that is not the main topic of the paragraph, or indeed of this paper. However, we are happy to add a reference to more recent advances in this field as summarised in Matthews et al. 2021. We changed the sentence in line 75 to

“for more recent developments see [5]”

Line 185 - Please check Freitas et al. 2019 (DOI:10.1038/s41598-019-51786) for updates on the acceptance of this biogeographical region

We are aware of this discussion but we think that this detail is not really relevant to the topic of our manuscript. We rephrased the sentence to:

“The Azores and the Canary Islands are two of the archipelagos that make up Macaronesia.”

Line 200 – “what relevant data is compiled from this paper? Not clear”

The paper mentioned is

Aranda, S.C.; Gabriel, R.; Borges, P.A.; Santos, A.M.; De Azevedo, E.B.; Patiño, J.; Hortal, J.; Lobo, J.M. Geographical, 558 Temporal and Environmental Determinants of Bryophyte Species Richness in the Macaronesian Islands. PloS One 559 2014, 9, e101786, doi:10.1371/journal.pone.0101786.

Indeed, the data used in this paper is also present in the other ones, so we removed it.

Line 206 – “How the taxonomic resolution used in this study may bias the results? Was the taxonomy updated before performing the models? Would they be different if considering subspecies as well? Did authors test this?

All the Calculations in Azores and Canary islands were made with the lowest taxonomic rank (species/subspecies). Therefore subspecies were already considered. We added the following sentence to the methods,

“Therefore, all calculated richness values for each island are for the sums of the lowest taxonomic rank (i.e. species or subspecies)”.

Line 215 – “I would exclude exotics from the total number of species, as their colonization processes are very different from the native ones.

Recalculate the results taking out introduced taxa not to bias results of total number of species and discuss the difference in results of including them.

We addressed this issue above

Line 269 – “use the same number of digits for all figures.”

Thank you for pointing this out. We have now the same number of digits after the decimal point for all figures, except for the integer numbers.

Line 269 -Detail all acronyms in the caption (e.g., Sd).

Thank you for pointing this out. We have now added the meaning of all acronyms in the caption of the Tables. We also added the following text (new lines 257-258):

“In addition, we also show the effective number of samples, “n_eff”, that is, the estimated number of independent samples [45].”

Line 273 – Table 3

Thank you for pointing out that the caption was incomplete.

We changed Table 3, it now looks like Tables 6, 7, 8 and 9. We also changed the captions of Tables 6, 7 and 8.

Around line 269 – Concerning Tables 1 and 2. “MERGE WITH TABLE ABOVE by including an extra line above the first describing the archipelago name (Azores/ Canaries)

We don’t agree with this recommendation because it would make the table too complicated. Therefore, unless there are editorial rules concerning this, we will keep the tables as they are now.

Lines 279 and 283, Tables 4 and 5. “what is the meaning of the grey shading? Please detail it in the caption

Thank you for pointing this out. We now added to the caption of Table 4:

“The grey shading areas highlight the three major groups of islands: the western, central and eastern groups.”

And to the table 5

“The grey shading areas highlight the two major groups of islands: the western and eastern groups.”

Line 288 – “?? Do you mean Plots (a and b)?” (and see also comment line 292.

We are not sure we understand this comment. We meant both plots, and we believe we are following the common format of this journal. Please let us know if we should change the format to something else.

Line 289 – “detail the acronym

We added Island Species-Area Relationship (ISAR)

 Lines 289 and 293 – “Not visible.”

We apologize for this, but we are not sure what happened because the shadows and the lines are visible in our original Word version.

Line 291 – “Captions need to be self-explanatory. If not Authors should clearly state were readers can find the information referred to in it.”

Thank you for pointing out this. We added after “c and z”:

 “of S=cAz”,

Line 305 – “tone down”

This sentence was removed.

Line 314 – “not visible at all in any of the islands. Please correct this

As we mentioned above, we apologize for this, but we are not sure what happened because the shadows and the lines are visible in our original Word version.

Line 320 – “This is methods, please restructure this

Thank you for this suggestion. Indeed, this paragraph repeats what had already been said in the “Methods”. We simplified it to:

“We now considered models with each of the explanatory variables, “area”, “elevation” and “distance to the mainland” separately and all together (we call the latter the “full” model). We fitted the models for different types of species: all species, endemic, SIE and MIE species.”

Line 336 – “same for the Canary Islands! Please make this clearer”

Indeed, we added to the end of this paragraph the following sentence:

“, and according to the DWAIC the four other models also have considerable support.”

Line 341 – “the distance models also have support. Make this clearer

Thank you for pointing this out. We rephrase the sentence to:

“but both the area, elevation and distance models also have…”

Line 342 Not really, as in both archipelagos elevation models are not significantly better than any other (as varWAIC is lower than 2)! Please rephrase.

We may not have explained clearly what we meant here. We rephrased the sentence to:

“As before, the models with “elevation” have a smaller DWAIC for the Canaries than for the  Azores”.

Line 346 – “delete”

We agree with this suggestion and deleted: “, DWAIC=2”

Line 347 - “replace by "lower WAIC"”

We agree with this suggestion and replaced “more support” by “lower WAIC”.

Line 348 - “this is discussion and should be deleted from here”

We agree and removed this sentence

Lines 352 – “and the fulle (both should be mentioned as both are not significantly different

and

Not true. Any of the models is with similar support and not significantly different from others. Rephrase

We changed the sentence to:

“Finally, for the MIE species (Table 9) and for the Azores, the models with more support are the area only and the “fulle” and for the Canaries the one with smallest WAIC is with elevation only model, although all the others also show significant support.”

Line 353 – “idem”

We rephrase the sentence to:

“Continuing the previous trend, the model with “elevation” only tends to exhibit lower DWAIC for the Canary Islands than for the Azores.”

Line 353 – “So, both use the exponential relationship? Not clear the difference between models. Rephrase”

Thank you for pointing out this mistake. We meant

“dist” and “full” through a power law relationship.”

We corrected it now.

Line 361 – “so you are not considering distance among islands, but distance to the mainland? Not clear. Rephrase.”

All the models with the Gaussian process included distance among islands. However, as we explained in the Methods in some models we also considered “distance to the mainland”.  To have this clear we changed the caption to:

“poisson” means that the model assumes a Poisson likelihood and “gp” means it included a Gaussian process (thus necessarily the distance among islands). “area”, “dist” and “elev” stand for the models that (in addition to the Gaussian process) also consider area, distance to the mainland or elevation, respectively, while “full” stands for a model with all these three explanatory variables included. The “e” in “diste” and “fulle” means that the distance to the mainland entered the model through an exponential relationship and “dist” and “full” through a power law relationship.”

Around lines 372, 376 – Concerning Tables 6,7,8. “MERGE WITH TABLE ABOVE”

See above. Therefore, unless there are editorial rules concerning this, we will keep the tables as they are now.

Line 414 – “this was already considered in other phylogeography works, including for the Canary Islands (e.g. Sanmartín, I., Van Der Mark, P., & Ronquist, F. (2008). Inferring dispersal: a Bayesian approach to phylogeny‐based island biogeography, with special reference to the Canary Islands. Journal of Biogeography, 35(3), 428-449.https://doi.org/10.1111/j.1365-2699.2008.01885.x)

Thank you for pointing out this reference, which we were not aware of (we added it to the references, ref. [48]). The sentence now reads:

“The novelty of our work, in the context of species richness in island biogeography, consists of also including the distance among the islands, and hence considering spatial autocorrelation using a Gaussian process (see also [48]).”

Line 433 – “That taxa richness is a function of habitat and elevation following the GDM has also been proven for other Macaronesian Archipelagos for other arthropods groups and flora and vertebrate groups. Provide some references to support this study and broaden the discussion, making it more interesting for a wider range of readers.”

We changed this paragraph to:

“However, an important insight from GDM theory is that topographical complexity determines species richness. In fact, there is a large number of theories arguing for a different range of determinants of species richness as a function of topographical isolation and elevation ([12][50,51]) and some arguing that “diversity begets diversity” [10] (but see [52]). Our work may add to this debate by providing a method to include spatial autocorrelation. Overall, elevation seems to have a more dominant role for the Canary Islands than it has for the Azores”

Which implied adding the following references

Borges, P.A.V.; Hortal, J. Time, Area and Isolation: Factors Driving the Diversification of Azorean Arthropods. J. Biogeogr. 2009, 36, 178–191, doi:10.1111/j.1365-2699.2008.01980.x.

Steinbauer, M.J.; Field, R.; Grytnes, J.-A.; Trigas, P.; Ah-Peng, C.; Attorre, F.; Birks, H.J.B.; Borges, P.A.V.; Cardoso, P.; Chou, C.-H.; et al. Topography-Driven Isolation, Speciation and a Global Increase of Endemism with Elevation. Glob. Ecol. Biogeogr. 2016, 25, 1097–1107, doi:https://doi.org/10.1111/geb.12469.

Pereira, H.M.; Proenca, V.M.; Vicente, L. Does Species Diversity Really Drive Speciation? ECOGRAPHY 2007, 30, 328–330, doi:10.1111/j.2007.0906-7590.04779.x.

Line 452 - “this is methods. Delete”

We followed this suggestion.

Line 456 – “Not always, as you have only tested two archipelagos and arthropod data. Soften the tone.”

It is likely that the meaning of our sentence was not correctly interpreted. We did not mean that spatial autocorrelation is always present, but that we should investigate if it is present.

The sentence now reads:

The existence of off-the-shelf methods to identify spatial autocorrelations, and as our results exemplified, warrants the conclusion that spatial autocorrelation should always be initially considered when studying the species richness of archipelagos, even if this analysis ends up showing that it is not present, which in itself would be an important conclusion.

We changed Figure 1, in order to make it more legible.

Reviewer 3 Report

Dear Authors: 

This research illustrated how to model an ISAR when considering the spatial autocorrelation. This analysis is very necessary in analyzing the relationship between the number of species and island area. I just have one major concern, please see below:

For the spatial autocorrelation among islands, my understanding is that the closer the two islands are in space, the more similar of species composition between islands (i.e., beta diversity). For alpha diversity (e.g., the species richness), however, how species richness on the island could be affected by spatial autocorrelation that needs further explanation. Therefore, before doing this research, the authors should first explain why we should consider spatial autocorrelation between islands when we analyze ISAR (It is better to provide a theoretical case). We also know that in many studies, the distance to the nearest island or the proportion of the island area within a certain range around the target island has been taken as a measurement of isolation, and considered the impact of isolation and island area on species diversity at the same time. In this study, it seems that the distance of the nearest island was taken as a covariate. I think the authors should also introduce how to distinguish the effect of spatial autocorrelation and isolation on the species diversity on islands. I feel that the introduction of the current manuscript needs to be further revised and improved.

Other minor comments

In Figure 2, the darkness of the lines between the islands represents the strength of the correlations in plots (c) and (d), but I can’t find the darkness of the lines.

Author Response

Reviewer 2

Dear Authors: 

This research illustrated how to model an ISAR when considering the spatial autocorrelation. This analysis is very necessary in analyzing the relationship between the number of species and island area. I just have one major concern, please see below:

For the spatial autocorrelation among islands, my understanding is that the closer the two islands are in space, the more similar of species composition between islands (i.e., beta diversity). For alpha diversity (e.g., the species richness), however, how species richness on the island could be affected by spatial autocorrelation that needs further explanation. Therefore, before doing this research, the authors should first explain why we should consider spatial autocorrelation between islands when we analyze ISAR (It is better to provide a theoretical case). We also know that in many studies, the distance to the nearest island or the proportion of the island area within a certain range around the target island has been taken as a measurement of isolation, and considered the impact of isolation and island area on species diversity at the same time. In this study, it seems that the distance of the nearest island was taken as a covariate. I think the authors should also introduce how to distinguish the effect of spatial autocorrelation and isolation on the species diversity on islands. I feel that the introduction of the current manuscript needs to be further revised and improved.

Thanks for this comment. In fact, we alluded to this question of mechanisms that may justify autocorrelation for alpha diversity in lines 83-87, which we repeat here:

“For instance, the distance among islands is likely to affect the ability of species to disperse within the archipelago [16–18] and closeness is likely to be a proxy for similar environmental conditions and for the islands’ age [16]. Therefore, when trying to fit an ISAR, the distance among islands should be incorporated into the model, or in other words, we should consider spatial autocorrelation.”

However, we do not necessarily agree with the following statement. “Therefore, before doing this research, the authors should first explain why we should consider spatial autocorrelation between islands”. Even if we do not know of any processes that may lead to spatial-autocorrelation, we argue that we should still consider it, since there may be unknown mechanisms leading to it. For instance, finding spatial autocorrelation where it was unexpected may lead to the search of (so far) hidden processes. Therefore, we prefer to start by assuming spatial autocorrelation and dismiss it if the analysis shows that it is largely irrelevant (but not a priori).

In order to ensure that this point is clear we add to the introduction the following text:

“For instance, the distance among islands is likely to affect the ability of species to disperse within the archipelago, which may affect not only species richness but also species composition (and beta-diversity) [16–18]. The latter is not the object of this work, however, closeness may affect species richness because it is likely to be a proxy for similar environmental conditions and for the islands’ age [16]. Moreover, some islands may have been connected during glacial periods, more likely to occur for nearby islands, and their present communities may still reflect a previous species’ richer community [2]. Therefore, when trying to fit an ISAR, the distance among islands should be incorporated into the model, or in other words, we should consider spatial autocorrelation.”

Other minor comments

In Figure 2, the darkness of the lines between the islands represents the strength of the correlations in plots (c) and (d), but I can’t find the darkness of the lines.

Thank you for pointing out this problem. We apologize for this, but we are not sure what happened because the shadows and the darkness of the lines are visible in our original Word version. We will do our best to ensure this won’t happen in the next manuscript version.
